

# Laboratory evaluation of different formulations of Stress Coat® for slime production in goldfish (*Carassius auratus*) and koi (*Cyprinus carpio*)

Raghunath B. Shivappa[1], Larry S. Christian[2], Jerry M. Law[3] and Gregory A. Lewbart[4]

[1] Pharmaceutical Development and Manufacturing Sciences, Janssen R&D (Pharmaceutical companies of Johnson & Johnson), Malvern, PA, United States of America
[2] Veterinary Services, North Carolina Museum of Natural Sciences, Raleigh, NC, United States of America
[3] Department of Population Health and Pathobiology, North Carolina State University, Raleigh, NC, United States of America
[4] Department of Clinical Sciences, North Carolina State University College of Veterinary Medicine, Raleigh, NC, United States of America

## ABSTRACT

A study was carried out to assess the effect of Stress Coat® on slime production in goldfish (*Carassius auratus)* and koi (*Cyprinus carpio*). The study also investigated histological changes that might be associated with slime producing cells, and wound healing in koi. Several formulations of Stress Coat® were investigated and the results showed that polyvinylpyrrolidone (PVP), also known as povidone, an ingredient of Stress Coat®, when used alone, showed significantly higher slime production in goldfish than salt and Stress Coat® without PVP after 25 h. The results also showed that koi treated with compounds containing PVP showed better wound healing than those not exposed to PVP. Histology results showed no difference between compounds tested with regards to density and number of slime producing cells.

## INTRODUCTION

The first line of defense for fish against parasites and other invading organisms is the epithelial mucus or "slime" coat. Both external and internal epithelial surfaces of fish are covered with a layer of mucus that provides protection against environmental factors like microorganisms, toxins, pollutants, acidic pH and hydrolytic enzymes (*Ræder et al., 2007*; *Gomez, Sunyer & Salinas, 2013*; *Guardiola et al., 2013*). Slime is produced on a continuous basis that allows for trapping of pathogens that can be washed into the water as mucus. Secretory mucins, produced by goblet cells (*Shephard, 1994*), are the major constituents of the mucus layer in which several biochemical compounds have been identified, including lysozyme (*Fletcher & White, 1973a*), antimicrobial peptides (*Cole, Weis & Diamond, 1997*) and antibodies (*Fletcher & White, 1973b*). For aquarium hobbyists it is very important that fishes are healthy and stress free to maintain good slime production. It has been reported

Corresponding author
Gregory A. Lewbart,
greg_lewbart@ncsu.edu

that stress causes chemical changes in mucus that decreases its effectiveness as a chemical barrier against invading organisms (*Francis-Floyd, 2002*; *Vastos et al., 2010*; *Ma, Huang & Want, 2013*). Stress upsets the normal electrolyte (sodium, potassium, and chloride) balance, resulting in excessive uptake of water by fresh water fish and dehydration in saltwater fish. Handling trauma physically removes mucus from the fish. This results in decreased chemical protection, decreased osmoregulatory function, decreased lubrication (causing the fish to use more energy to swim), and disruption of the physical barrier against invading organisms. Chemical stress (e.g., chemotherapeutant treatment) often damages mucus resulting in loss of osmoregulatory function, loss of lubrication, and damage to the physical barrier created by mucus (*Francis-Floyd, 2002*).

Several products are available in the ornamental fish industry claiming to maintain a healthy fish slime coat. Stress Coat®, a patented product of Mars Fishcare Inc., instantly removes chlorine and chloramines, making municipal tap water safe for fish (*Goldstein, Patel & Wiley, 1992*). The manufacturer claims it can also neutralize heavy metals and replace the natural slime coating that fish need in times of stress due to handling, shipping or tank mate aggression (there is no published evidence for these claims). The product contains aloe vera, "nature's liquid bandage," that can help prevent loss of essential electrolytes and protect damaged tissue against pathogens. Aloe vera has been reported to contain pharmacologically active ingredients advocated for the healing of several diseases in humans (*Ferreira et al., 2007*). One of the active ingredients in Stress Coat® is polyvinylpyrrolidone (PVP), a compound that has been widely tested and used in human and veterinary medicine as an effective wound healing accelerator and disinfectant when combined with iodine and other compounds (*Mayer & Tsapogas, 1993*; *Burks, 1998*; *Osborne & Yang, 1999*; *Kim et al., 2015*). Another common name for polyvinylpyrrolidone is povidone. The degree of effectiveness of povidone-iodine and wound healing is up for debate and results have been mixed (*Mayer & Tsapogas, 1993*; *Burks, 1998*). Stress Coat®, according to the manufacturer, may also may help heal torn fins and skin wounds and its use is recommended when setting up aquariums, changing water, or adding fish to a system.

In this study, we evaluated the efficacy of different formulations of Stress Coat® on slime production in goldfish (*Carassius auratus*) and koi (*Cyprinus carpio*). The specific objectives were to: (1) Quantify the amount of slime/mucus produced by goldfish and koi following administration of Stress Coat®, Compound "X" (PVP formulation only), Compound "Y" (Stress Coat® without PVP formulation) and saline (sodium chloride) at various time intervals. (2) Study histological changes in slime producing cells of koi exposed to the above compounds.

## MATERIALS AND METHODS

The study was carried out at the North Carolina State University College of Veterinary Medicine with the approval of the NC State Institutional Care and Use Committee Protocol #06-135-0. Effects of Stress Coat® and its different formulations on slime production were investigated in goldfish and koi. In experiments utilizing goldfish, slime produced was

calculated quantitatively by determining the weight of slime from a pre-determined area of skin. With koi, along with the weight of slime produced, 6 mm punch biopsies were obtained to evaluate histological changes, particularly, the number of slime producing cells. The experimental design is outlined below.

## Goldfish

Goldfish (with mean weight of 55.2 g) used in the study were purchased from a commercial fish hatchery and wholesaler in North Carolina. Fish were acclimated for 14 days and health status was evaluated with physical examination, behavior monitoring, and examination of gill, skin and fin biopsies. After acclimation, five goldfish per aquarium (four in the 25 h aquarium) were distributed randomly to 25, 60 L aquaria. The aquaria shared a common recirculating water system and water source. Environmental conditions were monitored and maintained within a narrow range (pH $= 6.9 - 7.6$; DO $= 7.4 - 8.6$ mg/L; ammonia $= 0.23 - 1.84$ mg/L NH3-N; nitrite $= 0.06 - 0.15$ mg/L NO2-N; nitrate $= 0.5 - 4.5$ mg/L NO3-N; temperature $= 21.2 - 21.6$ C). A 12-hour-light: 12-hour-dark cycle was maintained. Fish were fed a pelleted diet (Blue Ridge Fish Hatchery, Koi & Goldfish Food, All Season, Large Pellet Growth; Blue Ridge Fish Hatchery, Kernersville NC, USA) once daily (dependent on day & water quality parameters). All fish were healthy prior to and throughout the study.

Fish in the above-mentioned 25 aquaria were randomly assigned to four treatment groups and a control group that were sampled at five different intervals. Each fish within a tank served as a replicate (factorial design, $5 \times 5 \times 5$). Treatments were designated as Stress Coat® (at a dose of 10 ml/10 gallon), polyvinylpyrrolidone (PVP) only (at a dose of 10 ml/10 gallon), Stress Coat® w/o PVP formulation (at a dose of 10 ml/10 gallon) and (saline/salt at 3g/L). A control group that did not receive any chemical was also included in the study. All fish from each treatment group were sampled at 0 min, 15 min, 1 h, 4 h and 25 h. At every time interval, all fish from each treatment group were anaesthetized using buffered tricaine methanesulfonate (MS-222) at a dose of 200 mg/L, weighed, and slime was scraped from one 1 cm$^2$ area over the epaxial musculature using a preweighed plastic coverslip. The total weight of the cover slip along with the slime was recorded and the weight of slime was determined by subtracting the coverslip weight from the total weight.

## Koi

Scaleless koi (doitsu) (with a mean weight of 86.8 g) were purchased from the same vendor and acclimated under conditions similar to the goldfish. Three koi were randomly assigned to each of the four treatment and control groups in the same aquarium system described above. One koi from each treatment and control was sampled at 0 min, 1 h and 25 h time intervals. Slime weight was taken as described above on one side of the fish and the opposite side was used for the punch biopsy. A 6 mm punch biopsy was taken from a scale free area of each koi using a standard medical skin biopsy instrument (6.0 mm Sterile Disposable Biopsy Punch, Ref#: 33-36; Miltex, Inc., York, PA, USA) and the tissue was fixed in 10% neutral buffered formalin. The samples were then processed and stained using alcian blue to visualize slime producing cells and H&E to visualize thickness of skin layers. Biopsied

**Table 1** **Mean slime weight (mg) from goldfish treated with different compounds at various time intervals.** Values in bold are significantly lower ($P \leq 0.05$) then the time 0 values in the same row.

| Treatment | Time (hr) | | | | |
|---|---|---|---|---|---|
| | 0 | 0.25 | 1 | 4 | 25 |
| Salt | 11.2 | 8.6 | 5.4 | 7.0 | **2.0** |
| Stress Coat® w/o PVP | 6.2 | 6.8 | 8.0 | 6.4 | **3.3** |
| Stress Coat® | 8.8 | 8.6 | 9.2 | 9.2 | **4.8** |
| PVP only | 7.0 | 8.0 | 8.4 | 7.2 | 13.8 |
| Control | 9.2 | 7.6 | 6.0 | 6.0 | 7.5 |

**Table 2** **Mean slime weight (mg) from koi treated with different compounds at three time intervals.** Values in bold are significantly different ($P \leq 0.05$) then the other values in the same row.

| Treatment | Time (hr) | | |
|---|---|---|---|
| | 0 | 1 | 25 |
| Salt | 12.0 | **5.0** | **6.0** |
| Stress Coat® w/o PVP | 9.0 | 7.0 | 6.0 |
| Stress Coat® | 3.0 | 9.0 | 8.0 |
| PVP only | 10.0 | 9.0 | **3.0** |
| Control | 4.0 | 8.0 | **13.0** |

fish were monitored for 30 days to observe wound healing and potential wound related complications.

## Statistical analysis

Data from the study was analyzed using MINITAB 15 (State College, PA, USA). An ANOVA-General linear model was used to analyze the goldfish data. A two factor ANOVA without replication was used on weights of slime recorded from koi. A 95% confidence interval was used.

## RESULTS

### Slime weight
#### Goldfish

Mean weights of slime (mg) from goldfish belonging to different treatment and control groups at various time intervals are presented in Table 1. Goldfish treated with salt had significantly lower mucus weights at 25 h. Goldfish treated with PVP has significantly higher mucus weights at 25 h.

#### Koi

Mean slime weights (mg) from koi belonging to different experimental groups at three time intervals are presented in Table 2. Koi treated with salt and PVP had significantly lower mucus weights at 1 and 25 h. Control koi had significantly higher mucus at 25 h.

### Histological changes in koi

Koi from different experimental groups were biopsied at three time intervals. It was found that none of the treatment groups, except the control, showed a marked difference in the slime producing cells (Fig. 1). The control showed a higher number of slime producing cells at 1 and 25 h compared to 0 h. With regards to epidermal thickness (Fig. 2), The Stress Coat® group showed the highest level of epidermal thickness.

### Inflammation and secondary infection

The open biopsy sites in all koi showed clinical signs (hyperemia) of inflammation and secondary bacterial infection (Fig. 3).

### Wound healing

Wound healing in biopsied koi was monitored for a period of 2 weeks. Evaluation of wound healing was subjective and based on lesion color, size, and, depth. At the end of 2 weeks, it was determined that the three koi treated with salt and PVP remained healthy and showed a higher degree of healing than other treatment koi and the control group (Fig. 3).

## DISCUSSION

Slime weight was significantly lower ($P < 0.05$) in goldfish that were treated with salt for 25 h. One possibility for this would be the general decrease in stress that salt provides to freshwater fishes and with that stress reduction might come a decreased mucus and inflammatory response. When slime weight for various experimental groups was compared at each time interval, fish treated with PVP for 25 hr showed significantly higher ($P < 0.05$) mucus weight than fish treated with salt and Stress Coat® w/o PVP for 25 h (Table 1). The cause for this could be related to stress or inflammation initiated by the PVP (*Shephard, 1994*; *Vastos et al., 2010*). Although this finding may lead to speculation that PVP in Stress Coat® might increase mucus production after 25 h, more detailed studies are needed to confirm the effect since no significant difference ($P > 0.05$) was observed between PVP-only treated fish and controls. Overall, the findings indicate that the compounds used in the study may not play a significant role in increasing mucus production. However, extensive studies using techniques that evaluate the slime thickness accurately would likely yield conclusive results.

Koi were used primarily to assess the histological changes in slime producing cells. Hence, only one koi was sampled from each experimental group with no replication at three time intervals, which resulted in a small sample size, not worthy of statistical analysis.

Various physiological factors might be associated with the number of slime producing or goblet cells (*Shephard, 1994*). These include decreased oxygen levels (*Vastos et al., 2010*), increased nitrate (*Vastos et al., 2010*), social hierarchy stress (*Bonga, 1997*), and toxic metals in the water (*Bonga, 1997*). When epidermal layers (where slime producing cells are located) of koi were compared, koi treated with Stress Coat® showed higher thickness than other treatments (Fig. 2). However, the koi randomly assigned to this treatment group turned out to be relatively larger than those in the other treatment groups, which might have resulted in this observation.

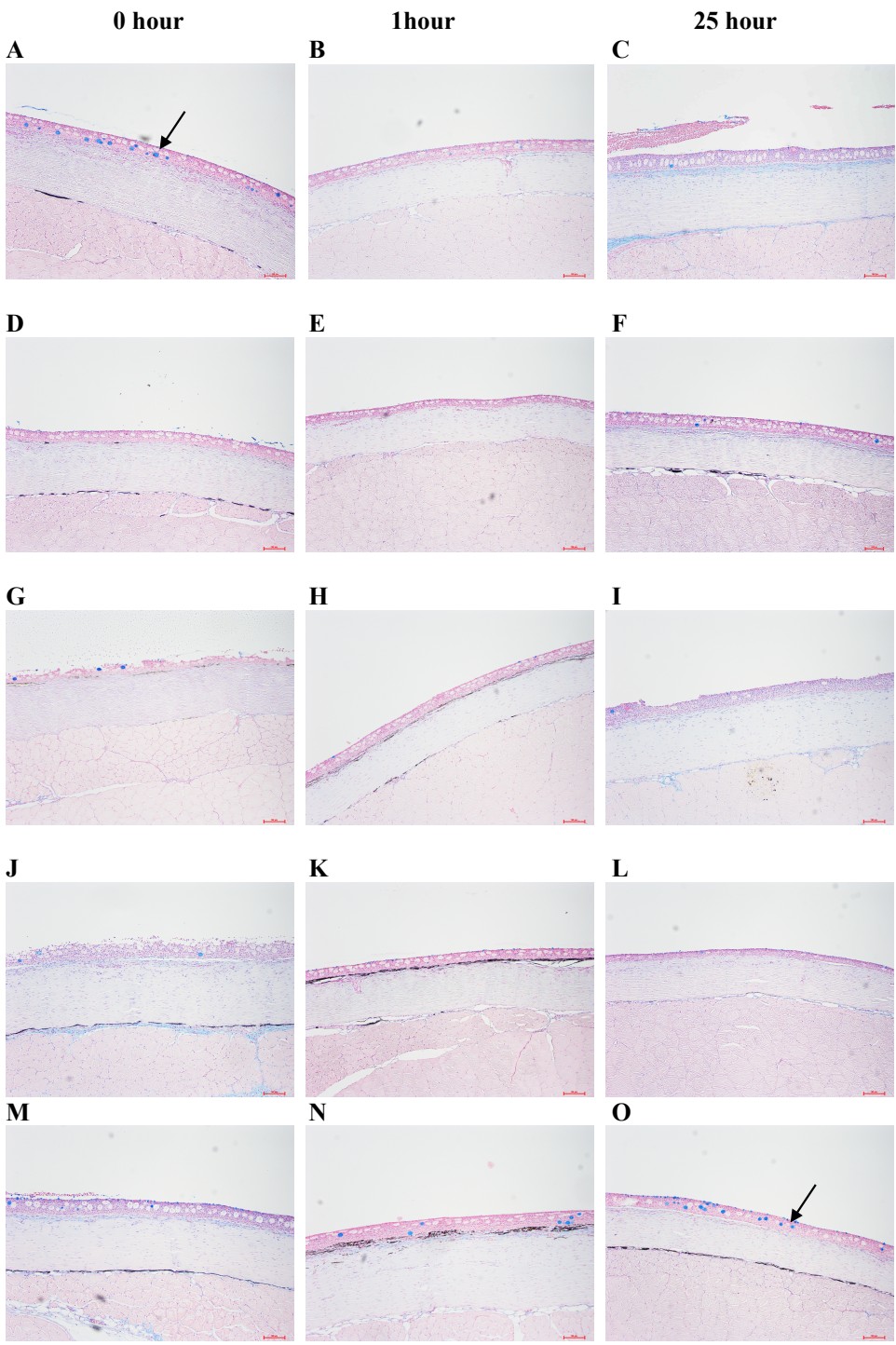

**Figure 1** **Histological changes in the number of slime producing cells (indicated by arrows) in koi treated with different Stress Coat® formulations, salt and control at different intervals.** (A–C) Stress Coat®; (D–F) Stress Coat® without PVP; (G–I) PVP only; (J–L) Salt; (M–O) Control. The control (M–O) showed a higher number of slime producing cells at 1 and 25 h compared to 0 h but none of the treatment groups showed any differences. The tissue sections were stained with alcian blue.

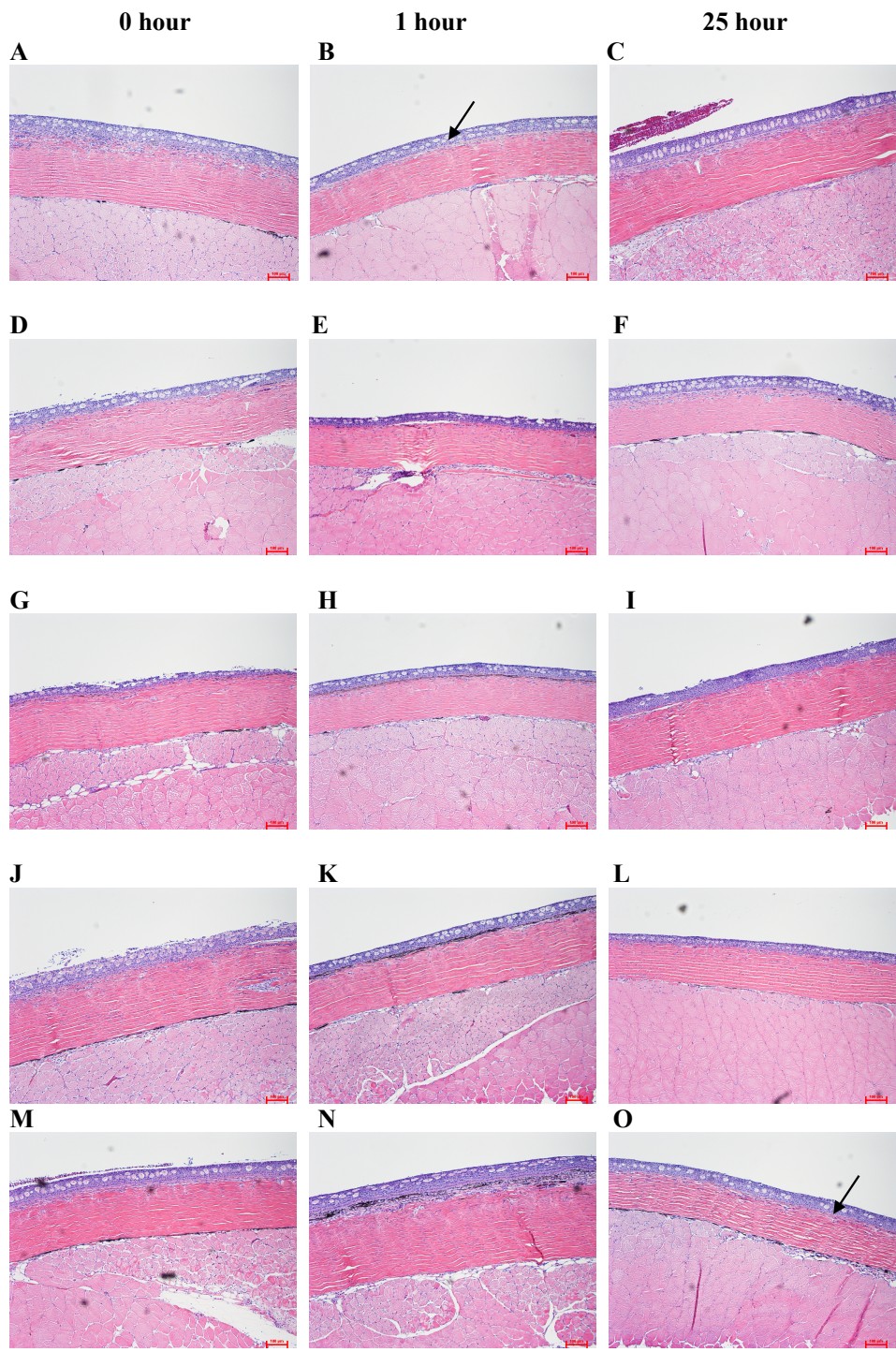

**Figure 2 Epidermal thickness (marked with arrow) from koi treated with different Stress Coat®**
**formulations, salt, and the control at different time intervals.** (A–C) Stress Coat®; (D–F) Stress Coat®
without PVP; (G–I) PVP only; (J–L) Salt; (M–O) Control. The koi treated with Stress Coat® showed
higher thickness than other treatments. The tissue sections were stained with H&E.

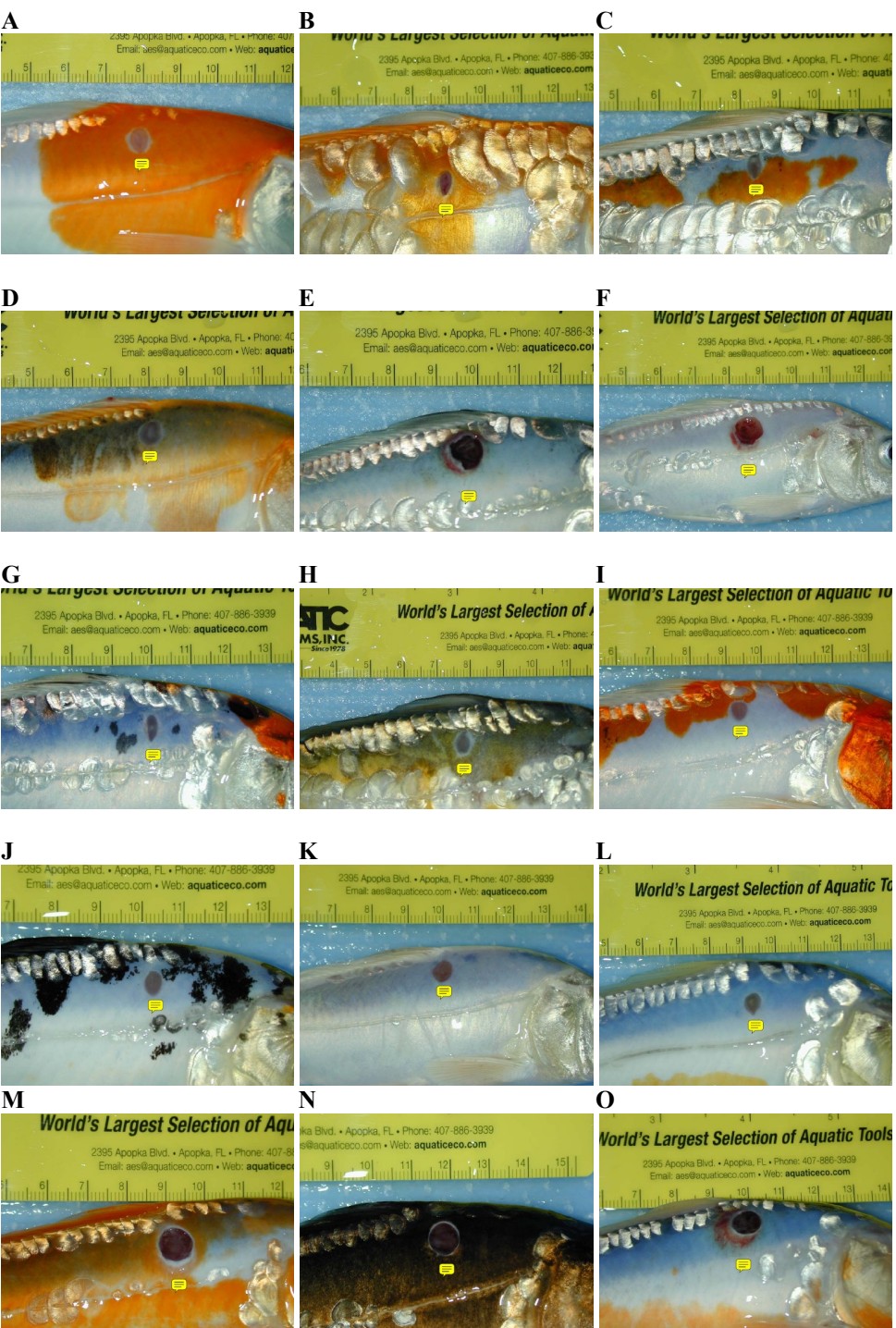

**Figure 3  Koi that were biopsied from all treatments were compared for wound healing at the end of 2 weeks.** Wound healing was evaluated subjectively and based on wound diameter, depth, and color. The small yellow text boxes have been inserted directly below the biopsy sites. (A–C) Stress Coat®; (D–F) Stress Coat® without PVP; (G–I) PVP only; (J–L) Salt; (M–O) Control. One can observe very good wound healing in the Stress Coat®, PVP, and salt rows. Healing was minimal in the control row and two of the three fish in the Stress Coat® without PVP row displayed little healing.

The control group, with the least amount of observed healing, suggests the compounds tested have some positive effect on wound healing. In addition to relatively poor healing, koi from the control group also developed what appeared to be secondary bacterial infections (presumptive). The observation appears to indicate that PVP in Stress Coat® contributes positively to wound healing, especially since the fish exposed to Stress Coat® without PVP showed lesser healing than Stress Coat® alone. Polyvinylpyrrolidone is known to speed healing in mammals (*Osborne & Yang, 1999*; *Kim et al., 2015*), and, the same appears to hold true for cyprinid fishes.

This work adds to our understanding of wound healing and mucus secretion in cyprinid fishes from an applied, clinical angle. Veterinarians and other health care professionals dealing with wounds in fish now have some evidenced based data to rely on when applying Stress Coat® and/or salt to the water containing goldfish and koi.

## CONCLUSIONS

The effect of Stress Coat® on slime production in goldfish was studied. The study also investigated histological changes that might be associated with the slime producing cells and wound healing in koi. It was found that no Stress Coat® treatments resulted in a significantly higher slime production during any of the time intervals in goldfish. However, slime production in PVP-only exposed goldfish was significantly higher than salt and Stress Coat® without PVP after 25 h. This gives an indication that PVP present in Stress Coat® is an important component that results in enhanced mucus production. This observation is further strengthened by the results of wound healing, where fish that were treated with compounds containing PVP showed better wound healing than those that did not receive PVP. Hence, it appears that PVP plays an important role in wound healing and mucus production. In koi, histology results showed no difference in slime producing cells in any of the treatment groups. Further studies with techniques that measure slime more accurately are recommended.

## ACKNOWLEDGEMENTS

The authors would like to thank Ms. Veronica Getty for technical help and Dr. Ratna Sharma-Shivappa for statistical analyses.

### Funding

This work was supported by Aquarium Pharmaceuticals, Inc., which is now Mars Fishcare Inc. The funders had no role in study design, data collection and analysis, decision to publish, or preparation of the manuscript.

### Grant Disclosures

The following grant information was disclosed by the authors:
Mars Fishcare Inc.

## Competing Interests

Dr. Shivappa is an employee of Novartis Inc., Holly Springs, North Carolina, United States.

## Author Contributions

- Raghunath B. Shivappa conceived and designed the experiments, performed the experiments, analyzed the data, wrote the paper, prepared figures and/or tables, reviewed drafts of the paper.
- Larry S. Christian conceived and designed the experiments, performed the experiments, analyzed the data, prepared figures and/or tables, reviewed drafts of the paper.
- Jerry M. Law conceived and designed the experiments, analyzed the data, contributed reagents/materials/analysis tools, wrote the paper, prepared figures and/or tables, reviewed drafts of the paper.
- Gregory A. Lewbart conceived and designed the experiments, performed the experiments, contributed reagents/materials/analysis tools, wrote the paper, reviewed drafts of the paper.

## Animal Ethics

The following information was supplied relating to ethical approvals (i.e., approving body and any reference numbers):

The study was carried out at the North Carolina State University College of Veterinary Medicine with the approval of the NC State Institutional Care and Use Committee: NC IACUC Protocol #06-135-0.

## Supplemental Information

Supplemental information for this article can be found online at http://dx.doi.org/10.7717/peerj.3759#supplemental-information.

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
