# Peer review of "Laboratory evaluation of different formulations of Stress Coat® for slime production in goldfish (Carassius auratus) and koi (Cyprinus carpio)"

_PeerJ, doi:10.7717/peerj.3759_

## Round 0.1 · original submission · Major Revisions

Though one reviewer recommended rejection of the manuscript, I would like to give a chance to make major revisions based on both reviewers' comments and suggestions. Please try to address all the points and resubmit to have better chance of being accepted.

Reviewer 1 ·

Basic reporting

A nice study and one that fills a hole in the literature about these synthetic slime coat products.

Please provide additional citations to support your statements about effects of stress on mucus quality and production, lines 60-69. While Francis-Floyd's extension publication mentions these factors, the article does not provide literature cited to follow up on these statements.

Lines 70-78 discuss the putative actions of Stress Coat related to fish handling and mucus replacement. It is important to emphasize that these are product claims. You do not cite and peer reviewed literature to support these claims, particularly as this relates to wound healing as mentioned in line 77. It would enhance these statements if such literature is available and could be cited.

Lines 135-140, please summarize the results for the goldfish and koi in each of these sections Rather than just referring the reader to the tables.

Lines 146-149, Was wound healing scored in some way? If so, please provide that scoring in tabular form if available. If not, a description of how the wound healing was assessed should be provided in the methods section.

Line 152, shouldn't this be Goldfish

lines 153-163. Can you speculate on why salt might decrease mucous production and some factors that might be involved with PVP associated increased mucous production?

Line 165, shouldn't this be Koi

Line 169 replace "worth of" with "worthy"

Line 170, can you describe some of the physiological factors that might be associated with the number of slime-producing cells?

Lines 175-177. please describe the evidence for secondary infections in the control group of koi, These findings should also be mentioned in the results section.

In your discussion, please speculate on how PVP might enhance mucus production and wound healing.

Figure 3 label: please how wound healing was assessed.

Tables 1 and 2 are a difficult to interpret. Consider another method to indicate significant differences. Perhaps color coding?

Figure 1 description: Remove "Office of Laboratory Animal Welfare". Consider the wording" Histological changes in the number of slime producing cells" in line 1. Remove the word "time" before the word "different" in line 2. Please summarize what is being illustrated in these images.

Figure 2 description: Remove "Office of Laboratory Animal Welfare". Please summarize what is being illustrated in these images.

Figure 3 description: Remove "Office of Laboratory Animal Welfare". Please summarize what is being illustrated in these images. Consider using something other than a small text box to identify the biopsy site, perhaps arrows. Please summarize/describe what is being illustrated in these images. Focus on a description of the indicators suggesting lack of healing the control group. It also appears that there is one row of images missing. Either the Stress Coat group or the control group.

Experimental design

Study meets the scope and times of the journal.

A brief explanation of how this research fills a knowledge gap would be useful.

Good experimental design for a preliminary study.

Please provide a disruption of how wound healing was assessed and compared.

Validity of the findings

If not in your discussion, please speculate on how PVP might enhance mucus production and wound healing.

Additional comments

This is a nice preliminary evaluation of one synthetic slime coat product commonly used in the fish handling and stress management.

The manuscript would have benefitted from a closer review prior to submission to ensure clarity of the presentation. That being said the study provides some nice support for the benefits of stress coat that have often been observed anecdotally.

Reviewer 2 ·

Basic reporting

The authors use clear English language. However, the structure of the manuscript suffers from several drawbacks.
a) The introduction lacks citations of relevant literature, especially concerning the described effectiveness of the Stresscoat product (line 70)
b) The authors hypothesize that the PVP contained in the Stresscoat product could be responsible for the beneficial effect of this product. However, no reason is given in the introduction to support this hypothesis. Also, no description of this chemical is done in the discussion together with possible hypothesis on its mechanisms of action.
c) The Results (line 133) are too cursory and just recall briefly the methods used. Some part of the discussion contains rough description of the results including statistical findings and should be therefore moved. Maybe the authors should fuse Results and Discussion.
d) In Figure 3: one line of panels is missing.
e) In the discussion, the paragraph entitled koi indeed describe the goldfish results...

Experimental design

The experimental design is well described except for the treatment used (line 106). The authors should state how long the treatment was applied to the fish. I expect the treatment to be on a limited duration of time as the fish are sharing the same recirculating system.

The investigation of the mucus production is done by weighting coverslip after scraping a small area on the fish. Likely this method is inducing high variation on the mucus collection. Concerning the goldfish, n=5 fish per treatment and per timepoint were sampled but nothing is indicated about the variability obtained (no standard deviations mentioned in Table 1, only means are shown in the supplementary raw data). Concerning the koi, n=1 were used, that appear to be too low to draw any conclusion.

Validity of the findings

To the opinion of the reviewer, the data obtained are not robust enough to allow the authors to conclude.
a) Concerning the mucus production. The conclusions raised by the authors seem contradictory throughout the manuscript. For example line 161 : “Overall, the findings indicate that the compounds used in the study may not play a significant role in increasing mucus production. “ compared to line 186 “This gives an indication that PVP present in Stress Coat® is an important component that results in enhanced mucus production”. To the opinion of the reviewer, the level of proof for this specific question is relatively low to conclude in any sense (mainly as a consequence of the method used). Also, only one timepoint show a significant difference between some groups. As stated by the authors, this needs probably more research with more precised ways to measure mucus production.
b) Concerning the histology, the authors state that no difference is found. However, they show one single slide per conditions. Probably, a more quantitative measurement such as the mean number of mucus productive cell per view found from different views and different fish would increase the robustness of the data.
c) Finally, the more interesting and robust finding is probably the better wound healing observed. However, one line of panel is missing in Figure 3. Also it could be interesting to follow the size of the lesion over time instead of focusing on a single time point.

Additional comments

In this manuscript, the authors describe the study of the effect of StressCoat on the mucus production, the number of mucus producing cells and the wound healing of goldfish and koi carp. Interestingly, they isolate one chemical compound as potentially responsible for the effect shown. These questions are practically relevant as they could help the fish farmer and hobbyist in how to use this StressCoat. Also it could help the further development of new products by isolating critical compounds. However, I do not believe that the manuscript is valuable enough to be published as such in PeerJ for the main reasons stated above. At least, the scientific writting and the riguor of the manuscript should be severly improved.

---

## Round 0.2 · accepted · Accept

I have confirmed that all the suggestions by the reviewers were well addressed. In my opinion, there is no need to add 'and wound healing in koi' in the title, rather the previous one is okay!

Thank you for addressing all the suggestions by the two reviewers.